# SARS-CoV-2-Specific Immune Responses in Vaccination and Infection during the Pandemic in 2020–2022

**DOI:** 10.3390/v16030446

**Published:** 2024-03-13

**Authors:** Wakana Inoue, Yuta Kimura, Shion Okamoto, Takuto Nogimori, Akane Sakaguchi-Mikami, Takuya Yamamoto, Yasuko Tsunetsugu-Yokota

**Affiliations:** 1Department of Medical Technology, School of Health Sciences and Graduate School of Medical Technology, Tokyo University of Technology, Tokyo 144-8535, Japan; g7123001be@edu.teu.ac.jp (W.I.); y_kimura@tmig.or.jp (Y.K.); g712300262@edu.teu.ac.jp (S.O.); amikami@stf.teu.ac.jp (A.S.-M.); 2Laboratory of Precision Immunology, Center for Intractable Diseases and ImmunoGenomics, National Institutes of Biomedical Innovation, Health and Nutrition, Osaka 567-0085, Japan; tnogimori@nibiohn.go.jp (T.N.); yamamotot2@nibiohn.go.jp (T.Y.); 3Laboratory of Aging and Immune Regulation, Graduate School of Pharmaceutical Sciences, Osaka University, Osaka 565-0871, Japan; 4Department of Virology and Immunology, Graduate School of Medicine, Osaka University, Osaka 565-0871, Japan; 5Research Institute, The World New Prosperity (WNP), Tokyo 169-0075, Japan

**Keywords:** SARS-CoV-2, COVID-19, infection and vaccination, serum and saliva, RBD and N-specific, IgG and IgA, T-cell responses

## Abstract

To gain insight into how immunity develops against SARS-CoV-2 from 2020 to 2022, we analyzed the immune response of a small group of university staff and students who were either infected or vaccinated. We investigated the levels of receptor-binding domain (RBD)-specific and nucleocapsid (N)-specific IgG and IgA antibodies in serum and saliva samples taken early (around 10 days after infection or vaccination) and later (around 1 month later), as well as N-specific T-cell responses. One patient who had been infected in 2020 developed serum RBD and N-specific IgG antibodies, but declined eight months later, then mRNA vaccination in 2021 produced a higher level of anti-RBD IgG than natural infection. In the vaccination of naïve individuals, vaccines induced anti-RBD IgG, but it declined after six months. A third vaccination boosted the IgG level again, albeit to a lower level than after the second. In 2022, when the Omicron variant became dominant, familial transmission occurred among vaccinated people. In infected individuals, the levels of serum anti-RBD IgG antibodies increased later, while anti-N IgG peaked earlier. The N-specific activated T cells expressing IFN γ or CD107a were detected only early. Although SARS-CoV-2-specific salivary IgA was undetectable, two individuals showed a temporary peak in RBD- and N-specific IgA antibodies in their saliva on the second day after infection. Our study, despite having a small sample size, revealed that SARS-CoV-2 infection triggers the expected immune responses against acute viral infections. Moreover, our findings suggest that the temporary mucosal immune responses induced early during infection may provide better protection than the currently available intramuscular vaccines.

## 1. Introduction

Since the outbreak of the first novel coronavirus caused severe acute respiratory syndrome (SARS-CoV) in Guandong, China, in November 2002 [1], another novel coronavirus emerged in Wuhan, China, in December 2019 [2,3] and rapidly caused a global pandemic. The virus, officially designated as SARS-CoV-2, is an enveloped single-stranded RNA virus belonging to a β-coronavirus family [4]. The SARS-CoV-2 infection occurred directly in the lung tissue through an angiotensin-converting enzyme (ACE)-II as a primary receptor [5], with the potential for development of severe pneumonia in especially the elderly and those with comorbidities. The disease caused by SARS-CoV-2 is called COVID-19.

The SARS-CoV-2 accumulated mutations continuously during human-to-human transmission and in chronic infections [6]. The WHO worked with the reported genetic mutation of the virus and assigned simple labels for key variants as variants of interest (VOIs) and variants of concern (VOC) in May 2021 (https://www.who.int/en/activities/tracking-SARS-CoV-2-variants/ (accessed on 29 January 2024)). From the virus arising from the Wuhan SARS-CoV-2 virus, Alpha (B.1.1 lineage) and Beta (B.1.35 lineage) variants were diverged, followed by the Delta (B.1.617 lineage) variant in October 2020 in India. At the end of 2021, the Omicron (B.1.1.529 lineage) variant was reported in South Africa and subsequently became a significant variant worldwide after the Delta variant. Although Omicron continues to expand as various sub-lineages, they have changed to preferably infect the upper respiratory tract (versus lower respiratory tract), as compared to pre-Omicron VOCs (https://www.who.int/news/item/16-03-2023-statement-on-the-update-of-who-s-working-definitions-and-tracking-system-for-sars-cov-2-variants-of-concern-and-variants-of-interest (accessed on 29 January 2024)) resulting in the attenuated phenotype. 

The advancement of novel vaccine technology appeared to help us achieve herd immunity against SARS-CoV-2 infection in the general population, at least initially. The COVID-19 vaccine was introduced in late 2020 and Watson et al. reported the global impact of the first year of COVID-19 vaccination through their mathematical modeling study [7]. In Japan, the mRNA-based vaccines, such as BNT162b2 (Pfizer/BioNTech) and mRNA-1273 (Moderna), as well as a defective adenovirus-based vaccine called ChAdOx1-S (Oxford), were introduced in 2021. Initially, the vaccination program was first provided to medical workers, but eligibility for free vaccines has since been extended to all age groups to achieve herd immunity (https://www.niid.go.jp/niid/ja/diseases/ka/corona-virus/2019-ncov/2484-idsc/10569-COVID19-53.html# (accessed on 29 January 2024)). 

However, with the surge of Delta variants, the decay of vaccine-induced neutralizing antibody response and the increase of SARS-CoV-2 reinfection have become of great concern, as seen in Israel [8]. It should be noted that, based on the experimental coronavirus infection study [9], the reinfection of human common-cold coronaviruses has been known to occur frequently. The COVID-19 Forecasting Team recently showed that past-infection-induced protection against re-infection from pre-omicron variants was very high [10]. However, the protection was substantially lower and shorter for the Omicron BA.1 variant [10]. Despite nationwide vaccinations, we have encountered eight epidemic peaks in Japan at the end of 2023 (https://www.niid.go.jp/niid/ja/basic-science/epidemi/12252-epi-2023-02.html (accessed on 29 January 2024)).

Thus, although the natural infection appears to link to a lower incidence of SARS-CoV-2 infection than mRNA primary series vaccination [11], the reinfection frequently occurs after vaccinations and even after natural infections, probably associated with continuous virus mutations [7,8,10,12]. To improve the vaccine effectiveness, a bivalent vaccine has been developed that represents the BA.4/5 lineage of the Omicron variant. Nevertheless, Omicron variants still continue to expand into various sub-lineages while maintaining an attenuated phenotype. A study on medical workers by Shrestha et al. found that the bivalent vaccine provided modest protection when the BQ lineages were dominant, but did not demonstrate effectiveness against the XBB lineages [13].

During the COVID-19 pandemic in early 2020, the Japanese government implemented strict measures to limit social activity. Schools and offices were closed, and people were required to stay at home. This strategy worked initially, but the second wave of the virus hit Japan in the summer of 2020. In order to understand the effectiveness of the vaccine, we monitored the immune status of university students. We found only one case of infection in our immediate surroundings. We began our vaccination study in the middle of 2021, which coincided with the beginning of the surge of Delta variants surge. The students in our university managed to remain relatively COVID-free during the early waves of epidemics including that of Delta variants in 2021 by avoiding close contact with others. However, with the wave of Omicron variants started in Japan in late 2021 (https://www.niid.go.jp/niid/ja/basic-science/epidemi/12252-epi-2023-02.html (accessed on 29 January 2024)), a significant number of students were infected, mostly by family members.

The main immune responses that protect against acute virus infections are neutralizing antibodies and virus-specific T-cells. However, there is still much we do not understand about the differing dynamics of IgG and IgA antibodies in blood and saliva following SARS-CoV-2 infection and vaccination, highlighting a critical area of research that needs to be explored more thoroughly. To gain insight into how immunity develops against SARS-CoV-2 during the pandemic period of 2020 to 2022, we analyzed humoral and cellular immune responses in vaccinated and infected individuals, as well as close contacts who lived with infected family members but remained PCR or antigen-test-negative. Our previous work showed that saliva IgA was induced early (around 7 days post-infection) after influenza virus infection and could help evaluate mucosal immune responses [14]. To better understand the mucosal immune response in SARS-CoV-2 infection, we also analyzed saliva IgA and IgG antibody responses in infected and their close contacts. Although the sample size is limited, the study will provide an overview of the immune responses of those who are naïve to a novel SARS-CoV-2 in a small community.

## 2. Materials and Methods

### 2.1. Subject and Sample Collection

In 2020, one family member of a staff member working in a hospital was infected with SARS-CoV-2 and samples were collected periodically. In 2021, university staff members recruited volunteers, including their families and students, before vaccination. The volunteers did not display any symptoms of COVID-19 during the vaccination study and, if infected with SARS-CoV-2, were asymptomatic. The vaccination schedule consisted of two shots, with the second shot administered three weeks after the first. A third vaccination was recommended six months after the second shot. We collected blood and saliva samples before (pre) and two weeks after (first, second, and third post-vaccination) from twenty-two vaccinated donors (mostly primed from June to August 2021), as shown in Table 1. The whole blood was centrifuged first to separate serum or plasma and cell pellets. By removing coagulant proteins from plasma, serum samples were prepared and frozen at −80 °C. We also obtained peripheral blood mononuclear cells (PBMCs) from the cell pellets (buffy coats) by sedimentation with a Ficoll-Hypaque density gradient (Sigma-Aldrich, St. Louis, MO, USA). These PBMCs were kept frozen at −140 °C until use.

Since February 2022, SARS-CoV-2 infection has spread among young students who received vaccination in 2021, the information of which is shown in Table 2. The samples from infected donors (17 donors) were collected early (10–14 days) and late (~1 month) post-infection. We collected the samples from individuals who were in close contact with infected families (PCR+) but test-negative and asymptomatic (10 donors) at two similar time points (early and late). Their information is shown in Table 3. To better understand the chronological study, we included a Appendix A that displays the study periods for each cohort and the pandemic waves of SARS-CoV-2 VOC sublineages in Japan from 2020 to 2022.

Additionally, a member of one of the authors’ family had a mild fever and sore throat, which was followed by her sister, who developed a high fever. Their infection was confirmed through an antigen test. In this specific case, we were able to collect saliva samples in the early days after the infection.

For saliva sample collection, SalivaBio (Salimetris, Carlsbad, CA, USA) was used and centrifuged at 1710 g for 15 min, and the supernatants were transferred to new tubes, as previously described [14]. All these samples were collected with written informed consent and kept in a −80 °C freezer until use. In a previous study, we stocked blood and saliva samples from volunteers in our division until before the influenza season 2019 [14]. We utilized some of them as pre-COVID-19 samples. The study followed the Helsinki Declaration and was approved by the ethical committee of the Tokyo University of Technology (No. E18HS-023).

### 2.2. Antigens 

The nucleocapsid (N) and spike (S) protein, especially RBD region, are major determinants of immune responses in SARS-CoV-2 infection and vaccination [15,16,17,18]. Therefore, we selected N and RBD proteins as antigens to analyze immune responses to SARS-CoV-2 in this study.

The N protein is relatively conserved among human β-coronaviruses in terms of amino acid sequences and structures [19]. We used frozen stock of SARS-CoV-1 N protein, which was produced in 2007, as described, before using the pET-SUMO system [20], because the N protein of SARS-CoV-2 was shown to have approximately 90% homology with SARS-CoV-1 [21]. Wen et al. recently showed that COVID-19 patient-derived monoclonal antibodies were cross-reactive to the N protein of SARS-CoV-1 but less-so to other human β-coronaviruses [22]. The Receptor binding domain (RBD) derived from SARS-CoV-2 Wuhan-1 spike antigen was obtained commercially (Sino Biological, cat. 40592-VNAH). We used these antigens for the ELISA test.

For the in vitro stimulation of PBMC, a codon-humanized SARS-CoV-2 N DNA with Strep-tag II peptide coding sequence (WSHPQFEK) at 3′ end was synthesized and cloned into an expression plasmid produced by GenScript Japan Co., Ltd. (Tokyo, Japan). The Expi293TM Expression System Kit (Thermo Fisher Scientific, Waltham, MA, USA) made recombinant N protein under the company’s instruction. Both cell supernatant and lysates were combined, and we purified SARS-CoV-2 N using Strep-Tactin^®^ Sepharose^®^ (IBA Lifesciences GmbH, Goettingen, Germany) using the protocol provided by the company.

### 2.3. Standard Antibodies

We obtained the RBD-specific IgG S309 (PMID: 32422645) by transfecting Expi293 cells with plasmids that express S309 IgG heavy chain and light chain using the 293fectin transfection reagent (Thermo Fisher Scientific). On day 5 after transfection, the cell supernatant was clarified by centrifugation and filtered. The S309 antibody was purified from the supernatant by HiTrap rProtein A FF Column (Cytiva, Tokyo, Japan) using the AKTA go (Cytiva). We also utilized these heavy and light chain genes to produce an RBD-specific IgA expression plasmid, as described previously [14]. We obtained the recombinant anti-RBD-IgA antibody by transfecting this plasmid as the IgG antibody described above. The culture supernatant was purified using Peptide M/agarose (InvivoGen Inc., San Diego, CA, USA). These RBD-specific IgG and IgA antibodies served as a standard to measure RBD-specific antibodies. To assess the level of N-specific IgG and IgA antibodies, we purified the patient’s serum at one month post-infection and used it as a standard.

### 2.4. ELISA

The amount of RBD or N-specific IgA and IgG in each sample was measured using the ELISA system described before [14]. In brief, a Nunc 96-well microtiter plate (Thermo Fisher Scientific) was coated with an RBD-Spike or N protein at 1 μg/mL in PBS and kept at 4 °C overnight. The plate was washed and blocked with PBS/0.5% BSA at RT for 1.5 h, and the samples were added to the plate. Then, the plate was incubated for 1 h with a biotinylated anti-human IgG or IgA antibody (Southern Bio-Tech, Birmingham, UK), followed by HRP–streptavidin (1:2000 dilution with PBST, BioLegend, San Diego, CA, USA) for 30 min. Finally, the TMB substrate (Sigma-Aldrich) was added, and the color development was measured at OD450 using a microplate reader iMARKTM (BioRad, Hercules, CA, USA).

The amount of total IgA in saliva was measured as described previously [14]. As a standard, purified myeloma IgA1 protein (Sigma-Aldrich) was used. Note that the amount of total IgA was determined for each saliva sample, and antigen-specific IgA titer was normalized with total IgA, as previously described [14].

### 2.5. In Vitro Stimulation and Flow Cytometry

Frozen PBMCs were thawed, and 1 million cells were distributed to wells of a 96-flat-bottom culture plate (Corning Inc., Corning, NY, USA) in 0.1 mL of RPMI-1640 medium supplemented with 10% fetal bovine serum (FBS), 1/100 volume of Glutamax and Penicillin/Streptomycin (Thermo Fisher Scientific). The medium only or the medium containing 10 mg/mL of purified SARS-N antigen protein was added to make a final volume of 0.2 mL/well. A well containing SEB (*Streptococcus* enterotoxin B: Sigma-Aldrich) antigen at 1 mg/mL was prepared for each donor sample as a positive control. The plate was placed in a CO_2_ incubator (5% CO_2_) at 37 °C overnight. Sixteen hours later, brefeldin A (BFA) and monensin (both from Sigma-Aldrich) were added at a final concentration of 5 µg/mL and 2 µM, respectively. Simultaneously, 1 mL of PE-conjugated anti-CD107a antibody (BioLegend, San Diego, CA, USA) was added to each well, and the plate was incubated further for 5 hr.

Cells were collected into 5 mL polystyrene round tubes (Falcon^®^352008, Corning), washed once with PBS, reacted with LIVE/DEAD™ Fixable Aqua Dead Cell Stain (Thermo Fisher Scientific) for 15 min at room temperature (RT), and then, incubated for 20 min on ice with the following antibody mixture: FITC-conjugated anti-human CD69, PerCP-Cy5.5-conjugated anti-human CD8, PE-Cy7-conjugated anti-human CD45RA, APC-Cy7-conjugated anti-human CD27, BV421-conjugated anti-human CD3 and FcReceptor blocker. For intracellular staining, cells were washed twice with PBS and fixed by 100 µL/sample of fixation buffer supplied in eBioscience™ Intracellular Fixation and Permeabilization kit (Thermo Fisher Scientific) for 20 min on ice. These cells were washed twice with 500 µL/sample of permeabilization buffer in the kit and incubated with APC-conjugated anti-human IFN-γ. APC-anti-mouse IgG_1_ was used as an isotype control. After 20 min incubation on ice, cells were washed with PBS/2%FBS/0.05%NaN_3_ (SB) and resuspended in SB for flow cytometer analysis. All these mouse monoclonal antibodies were purchased from BioLegend.

Finally, stained cells were acquired using FACSVerse™ Cell Analyzer (Becton Dickinson, and Co. (BD), Franklin Lakes, NJ, USA) and re-analyzed by a Flowjo ver 10.8.1 (BD). After removing doublets, live and CD3-positive T-cells were gated for analysis. Then, the CD8^+^ and CD8^−^ T-cells were separately analyzed.

### 2.6. Statistical Analysis

The amount of SARS-CoV-2 specific IgA and IgG antibodies was calculated based on standard IgG and IgA antibody titers using Microplate Manager 6 (BioRad). Statistical analysis was performed using GraphPad Prism version 10.1.1. The differences between groups were analyzed using the Non-parametric *t*-test of the Wilcoxon matched-pairs signed rank test. A *p*-value < 0.05 * or <0.01 ** was considered statistically significant. 

## 3. Results

### 3.1. Immune Responses to SARS-CoV-2 Infection at Early Phase of Pandemic, 2020

Compared to European countries, our university had minor infections in 2020. However, we had an opportunity to analyze the serum samples of one medical doctor infected with SARS-CoV-2 in the hospital in August 2020. He had a slight fever for only half a day, while other symptoms were mild. Serum samples were collected consecutively at day 14, followed by one, four, and eight months post-infection. The serum taken in October 2019 for the influenza vaccine study was used as a pre-infection serum. We measured anti-RBD and anti-N IgG antibodies using an in-house ELISA system. As shown in Figure 1a, both the anti-RBD and anti-N IgG levels increased after infection and peaked one-month post-infection (p.i.). The levels of these IgG gradually declined but remained high at eight months p.i. At eight months p.i., he received the first COVID-19 mRNA vaccine, followed by a booster three weeks later. As expected, the level of anti-RBD IgG was augmented at nine months p.i. and slightly increased by the booster vaccination at ten months p.i. On the other hand, the level of anti-N decreased to the almost basal level at nine months p.i.

We conducted an analysis of T-cell responses to SARS-CoV-2 infection. We stimulated PBMCs with a purified recombinant SARS-CoV-2 N protein in vitro. After 18 hr of stimulation, we studied IFN-γ and CD107a production in activated (CD69^+^) T-cells with flow cytometry. The representative results of flow cytometry, taken one month after infection are shown in Figure 1b. The data at four-time points (pre, 1, 4, and 8 months) are depicted in Figure 1c. We observed that the percentage of N-specific CD8^+^ T-cells producing IFN-γ^+^ and CD107a^+^ cells was high at one month p.i. Although the rates of CD107a^+^ cells are relatively high even without antigen, the net increase of percentages is still at a peak at one month after infection. The results suggest that cytotoxic T-cells are highly activated at an early stage of infection. Based on these results, it is likely that cytotoxic T-cells become active at an early stage of the disease. Therefore, it may be necessary to analyze T-cell responses as early as possible.

### 3.2. The Vaccine-Induced Anti-SARS-CoV-2 RBD-Specific Antibodies in 2021

When the country-wide vaccination started in early 2021, we called for healthy volunteers who will receive COVID-19 vaccines in our department of Medical Technology. The serum samples at five time points, before immunization (pre), 10 days after priming (V1), 10–14 days after the second booster (V2), 6 months after the second (6M post-V2), and 10–14 days after 3rd immunization (V3), were obtained from 22 staff or students. The sample information is shown in Table 1. All these volunteers did not develop any COVID-19 symptoms during the vaccination study even when the Delta variant surged in Japan. The anti-RBD IgG and IgA levels are summarized in Figure 2a,b, respectively. After the first priming (V1), anti-RBD IgG slightly increased, and the level was augmented after the second booster (V2). At six months post-V2 vaccination, the level of anti-RBD IgG decreased to a basal level. Although the IgG level re-increased after the third vaccination (V3), the level was significantly lower than that of V2 (*p* < 0.05). As reported by Keshavarz et al. [23] the difference in the induced levels of anti-RBD IgG between vaccine BNT162b2 (Pfizer) and mRNA-1273 (Moderna) was not observed after the second vaccination (V2). Therefore, we combined both groups in V2. 

This trend was also observed in the level of anti-RBD IgA. However, because the level of anti-RBD IgA was 100-fold lower than that of anti-RBD IgG, the difference between V2 and V3 was insignificant. Note that the sample collection for the vaccination study started in 2021 just before the delta lineage virus became a significant lineage in Japan (https://www.niid.go.jp/niid/ja/basic-science/epidemi/12252-epi-2023-02.html (accessed on 29 January 2024)) and only a few students were infected in our university.

### 3.3. Serum Antibody Responses to SARS-CoV-2 in Infected and Close Contact Individuals

By accumulating mutations of SARS-CoV-2, a highly transmissible Omicron VOC detected in November 2021 became dominant in 2022, and permitted escape from infection- or vaccine-induced immunity [24]. On the other hand, attenuated replication and pathogenicity of omicron have been noted [25]. Since early 2022, we had many COVID-19 cases among students. We collected serum and saliva samples from 17 of them. They were all vaccinated in 2021 at least twice. Case C22-3 was unique because he was initially infected in the middle of 2021, before the vaccination started, and re-infected in April 2022 after immunization. The information on their infection/vaccination status is summarized in Table 2. We took samples early, within 14 days after infection (day 0 was defined based on the fever followed by PCR positivity), and at late, 1 month p.i. We also used serum samples stocked before the end of 2019 as pre-COVID-19 samples and measured the levels of anti-RBD and anti-N IgG in the serum with ELISA.

Serum anti-RBD and anti-N IgG were upregulated after infection. Anti-RBD IgG increased early, and the level was further increased later (Figure 3a, left). Because donors are all vaccinated, a higher level of anti-RBD IgG early may not necessarily reflect the infection. However, the further increase of IgG levels later indicated that the anti-RBD IgG response was augmented by infection. In contrast, serum anti-N IgG increased only early by infection, and a further increase was not observed later (Figure 3a, right).

While Omicron was more transmissible than Delta, it exhibited reduced disease severity in the period it co-existed with Delta [6]. Infection with the Omicron variant led to more asymptomatic cases than before, and new infections frequently occurred at home among family members. We collected ten samples of individuals with family members who had PCR-positive COVID-19 cases but remained uninfected. These close contacts were asymptomatic and coronavirus test negative. The information of these donors is shown in Table 3, and the levels of serum anti-RBD and anti-N IgG are depicted in Figure 3b. A significant variation in the level of anti-RBD IgG was observed (Figure 3a, left), and only a late increase of anti-RBD IgG was statistically significant (Figure 3b, left). However, anti-N IgG did not increase (Figure 3b, right). Because they were all vaccinated in 2021, we assumed their anti-RBD antibody response was boosted due to close contact with the infected. Asymptomatic contact individuals may be exposed to a limited amount of the virus transiently, but their immune systems are competent enough to eliminate it.

### 3.4. T-Cell Responses against SARS-CoV-2 in Infected Individuals

The T-cell responses against Spike and Nucleocapsid develop after SARS-CoV-2 infection or vaccination and are believed to be necessary for the protection [26,27,28]. We obtained PBMCs from seven infected donors at around ten days after infection (early) and from four infected donors over three months after infection (late) (Table 2). We also recruited four additional donors (donors in Table 1) who were infected at the end of 2022, and their blood was taken 10 to 14 days after infection. Because they are all vaccinated, we used a N antigen for in vitro stimulation. PBMCs were cultivated in the absence (no Ag) or in the presence of SARS-CoV-2 N antigen (SARS-N) overnight, then frequencies of activated (CD69^+^) and IFN- γ or CD107a positive CD4^+^ or CD8 T^+^ cells were analyzed by flow cytometry. 

As shown in Figure 4a, the SARS-CoV-2 N specific IFN-γ responses in CD8^+^ (left panel) and CD4^+^ (right panel) T-cells were variably detected, though the background level (no Ag) was high in some donors. The positivity of 11 samples collected at early infection was significantly increased in the presence of SARS-CoV-2 N antigen; *p* = 0.0186 in CD8^+^, but not in CD4^+^ T-cells (*p* = 0.0830). Although the statistical evaluation of four samples collected much later after infection is not feasible, N-specific IFN-γ^+^ memory T-cells will likely become marginal in both CD8^+^ and CD4^+^ T-cells.

As shown in Figure 4b, the frequency of CD107a^+^ T-cells also significantly increased at early infection; *p* = 0.0273 in CD8^+^ and *p* = 0.0068 in CD4^+^ T-cells. As regards the four donors at the late phase of infection, N-specific CD107a^+^ CD4^+^ T-cells with cytotoxic granules may have a trend to be maintained longer than N-specific CD107a^+^ CD8^+^ T-cells (Figure 4b, compare right to left panel).

Taken together, SARS-CoV-2-infected individuals can develop substantial SARS-N-specific CD8^+^ and CD4^+^ T-cells early after infection, but the longevity of SARS-N-specific memory T-cells needs to be studied further.

### 3.5. Saliva Antibody Responses in Infected and Close Contact Individuals

Saliva IgA represents upper mucosal immune responses. In influenza infection, virion-specific IgA antibody in saliva was mainly detected early at day 7–10 p.i. In comparison, in saliva, systemically induced IgG increased later at 3 to 4 weeks [14]. We collected saliva samples from infected and close contacts to understand the mucosal immune response in SARS-CoV-2 infection. However, salivary glands are known to be a reservoir for SARS-CoV-2 [29]. The saliva is a valuable specimen comparable to a nasopharyngeal swab sample for detecting SARS-CoV-2 in COVID-19 [30,31]. However, it was not feasible for non-medical individuals to consecutively collect saliva samples soon after infection for antibody analysis due to the risk of transmission. Additionally, the initial university regulations for COVID-19 mandated a two-week quarantine period for students before returning to campus. Therefore, saliva samples were collected simultaneously with serum samples at more than 14 days p.i. as before, followed by more than 1 month p.i. as late. As shown in Figure 5, no significant level of RBD nor N-specific IgA antibodies were detected in both infected (Figure 5a) and close contacts (Figure 5c). 

Concerning saliva IgG antibody, a significant level of anti-RBD IgG was detectable compared to pre-COVID-19 in both infected and close contacts, but no further increase was noted later (Figure 5b,d, left). Interestingly, the increase of saliva anti-RBD IgG was more evident in close contacts (Figure 5d, left) than in infected (Figure 5b, left). The results suggest that the anti-RBD IgG in saliva is derived from systemically induced IgG by vaccination, which may, to some extent, contribute to the protection in these close contacts. Saliva anti-N IgG was detected in neither of these groups (Figure 5b,d, right panels).

### 3.6. Early Kinetics of Saliva Antibody Responses in Infected Individuals

Thus, our collected saliva samples may not be suitable for analyzing early mucosal antibody responses in SARS-CoV-2 infection. The advantage of saliva is that saliva sampling has self-collection capabilities. When the Omicron raged in Japan, one of the authors was infected with SARS-CoV-2, and she developed a mild fever and sore throat (a). The next day, her younger sister developed a high fever (b). They were confirmed to be SARS-CoV-2 infected. Their saliva samples were collected consecutively every 2 days, starting from the day with fever (as day 0), 2, 5, 7, 9, and later. These saliva samples were heat-inactivated and analyzed for anti-RBD and anti-N antibodies with ELISA. The result is shown in Figure 6. In the first case, both anti-RBD and anti-N IgA antibodies peaked on day 2, and the second peak of anti-RBD IgA was observed on day 9 (Figure 6a). Also, in the second case, anti-RBD as well as anti-N IgA transiently increased on day 2. The second peak of anti-N IgA was observed on day 9. However, at a lower level than the first case (Figure 6b). In the second case, anti-N IgG increased on day 9, and anti-RBD IgG became high on day 14, whereas in the first case, both anti-RBD and anti-N IgG were only slightly increased 1 month later. Thus, mucosal IgA response was induced very early after infection, and IgG response appeared to follow much later after infection. The result suggests that we can obtain more definitive information only if self-collection of saliva on day two post-infection is feasible in all infected and close contacts.

## 4. Discussion

The COVID-19 pandemic led to changes in our way of life, but it also presented an opportunity to study how the human immune system responds to yet unencountered viruses. To gain insight into how immunity develops against SARS-CoV-2 from 2020 to 2022 when the virus evolved to escape from human immunity, we conducted a chronological analysis of immune responses in individuals who were infected with SARS-CoV-2, vaccinated against it, and later experienced post-vaccination infection along with the surge of Delta and Omicron variants. Our sample size is limited. Additionally, using only RBD and N proteins as assay antigens may not fully elucidate vaccination or infection status, and substantial levels of cross-reactivity have to be taken into consideration. The sample size is limited. Our study would provide valuable insights into how we, as humans, should combat newly emerged acute virus infections.

In the study of a single first case of COVID-19 infection in our surroundings, we observed a typical antibody response against acute virus infection that peaked at one month, gradually declining as expected. Anti-N IgG returned to near the basal level at nine months. Feng et al. reported that RBD- and full-length spike-IgG decreased during the first six months but remained stable up to one year after hospital discharge, then declined [26]. In our case, the earlier decay of anti-N IgG could be due to the disease severity, which is very mild without any pulmonary damage. By vaccination at eight months after infection, the level of anti-RBD IgG became 1-log higher than the peak achieved by infection. Thus, vaccination can significantly enhance the antibody response elicited by SARS-CoV-2 infection. The long-term follow-up study by Koerber et al. showed that virus-neutralizing antibody titers rapidly declined in convalescents after asymptomatic or mild SARS-CoV-2 infection over nine months. Still, vaccination enhanced both antibody and cellular responses, especially against SARS-CoV-2 spike [27]. Unfortunately, because of the lack of RBD protein for in vitro stimulation, we followed only N-specific T-cell responses starting from one month after infection. We detected activated (CD69^+^) IFN-γ^+^ and CD107a^+^CD8^+^ T-cells only transiently at one month. New technology has been developed to analyze T-cell responses during the SARS-CoV-2 pandemic [27,28,32]. This technology combines epitope megapool with multiparameter flow cytometry detection of novel activation markers. While our research detected relatively weak T-cell memory responses one month after infection, these technologies may help improve our assay sensitivity and detect such responses earlier. 

The level of anti-Spike IgG, particularly anti-RBD IgG, is important for protection against COVID-19, as it positively correlates with serum neutralizing capacity [33]. To assess the effectiveness of mRNA vaccines coding for the Spike gene, we analyzed the level of anti-RBD IgG and IgA in uninfected individuals’ serum. When healthy students were recruited for this study, following the second booster, both IgG and IgA levels in vaccinees significantly increased as expected. However, the IgG level gradually decreased over time, as reported earlier [27]. With the emergence of the delta variant, a third booster vaccination has been introduced [34,35,36]. In our cohort, the level of IgG increased again after the third vaccination but did not increase beyond the level seen after the second vaccination, which was also noted in a study by Kesharvarz et al. [23]. The difference in IgG peak levels between the second and third doses could be influenced by factors such as the age distribution of vaccine recipients, the timing of sampling, and the method of measurement. Interestingly, recent studies demonstrated that the IgG4 subclass antibody was upregulated in repeated vaccination of SARS-CoV-2 mRNA [37,38], indicating the IgG4 class switch mechanism for immune regulation [39,40]. Nevertheless, it is important to note that vaccine effectiveness against Delta variants increased significantly after the third vaccination [34,35,36,41].

It has been observed that cellular immune responses, specifically the spike-specific CD8^+^ T-cell responses that were produced by the original vaccines, are still very effective in responding to the Omicron variant [42]. This highlights the significance of T-cell responses in providing protection. During our study, we detected low but significant N-specific CD4^+^ and CD8^+^ T-cell responses two weeks after infection, but these responses were not detected 3–5 months later. Due to the small number of samples and limitations in our methodology, we were unable to analyze the longevity of T-cell responses. However, T-cells triggered by vaccination or infection against SARS can help in reducing the severity of the disease, as previously reported [42].

Reinfection of SARS-CoV-2 is becoming more common even after vaccination, particularly with the emergence of the Omicron variant. One reason is because the vaccine-elicited neutralizing antibodies have been greatly reduced [43]. Additionally, the Omicron variant has reduced virus pathogenicity [25], making reinfection or asymptomatic infection more likely than before [6]. In our study, all infected individuals were previously vaccinated with original mRNA vaccines in 2021, and their disease was mild. It is unknown whether the vaccine-induced anti-RBD IgG level remained or decayed just before infection. Note that our antibody assay samples measured anti-RBD, but not anti-Spike antibodies that may cover more epitopes than anti-RBD, and the RBD antigen used was derived from the sequence of the Wuhan strain. As a result, our assay of infected/close contacted individuals in 2022 may not be sensitive enough, and the observed antibody levels were mostly due to previous vaccinations. However, on the other hand, the increased level of serum anti-N IgG indicates the SARS-CoV-2 infection [15,18]. In our study, anti-N IgG increased early but was not augmented later, suggesting the transient nature of virus invasion. Close contacts of infected individuals did not show an increase in anti-N IgG, but their anti-RBD IgG levels were higher than pre-pandemic levels, likely due to previous mRNA vaccination. An interesting finding in the study was that close contacts who did not have detectable infection or antibodies gained T-cell immunity against SARS-CoV-2 [44]. This was determined by stimulating PBMCs in vitro with peptide pools for ten days. The authors suggested that exposure to the virus led to T-cell immunity even without a successful infection. However, more research is needed to determine if these T-cells can provide protective immunity. Alternatively, early mucosal IgA responses may have contributed to the inhibition of spreading infection in these close contacts.

Thus, many studies highlight the importance of vaccination as a critical strategy, at least initially, in the fight against COVID-19. However, the level and duration of antibody response may not be high enough. A study conducted by Wei et al. indicated that the protection against infection would last for 5–8 months after two BNT162b2 doses without prior infection, compared to 1–2 years in unvaccinated individuals after natural infection [45]. Additionally, almost all symptoms were reported less frequently in infected and vaccinated individuals than in infected unvaccinated individuals. The vaccinated were also more likely to be completely asymptomatic [46]. Carazo et al. showed in their study of healthcare workers that those who had two doses of mRNA vaccine and previous Omicron infection were better protected from re-infection compared to a third vaccine dose without infection, indicating the limited benefit from additional vaccine doses for people with hybrid immunity [47]. Moreover, as Omicron variants continue to accumulate mutations, it was recently reported by Shrestha et al. that the second mRNA vaccine developed to target BA.4/5 sublineage of Omicron variants had significantly reduced effectiveness, and it was not effective against the XBB lineages when they were dominant [13]. These results suggest the need for a more effective vaccine that can block the early invasion of the virus through the upper respiratory mucosa by inducing strong mucosal IgA immune responses.

Finally, we were unable to find any N-specific IgA antibodies in the saliva samples. The levels of anti-RBD IgG found in the saliva could result from the systemic IgG produced after vaccination [48]. In a previous study of influenza infection, we detected the presence of anti-flu IgA antibodies in saliva after 7 to 10 days of infection [14]. Moreover, Shan et al. reported that N protein presents very early (day 1–7 p.i.) in blood and saliva in SARS-CoV-2 infection [49]. Therefore, more than two weeks after the SARS-CoV-2 infection, the saliva sample collection appeared unsuitable for detecting early IgA response. Recently, Thomas et al. developed and evaluated salivary antibody assays for recent Omicron variants in order to detect previous SARS-CoV-2 infections [50]. They reported that the sensitivity of assays for ani-Spike was greater than that for RBD and N proteins. Nevertheless, in our saliva samples collected soon after infection, we were able to detect both anti-RBD and anti-N antibodies peaked at day two, followed by a second peak around day nine p.i. Our results suggest that the initial IgA-dominant mucosal response is weak and transient, whereas secondary memory responses involving systemic IgG are active after day nine p.i. Thus, if the self-collection of saliva samples is feasible, saliva will be more beneficial for detecting SARS-CoV-2 infection at mucosal sites. This may help us to understand how close contacts exposed to the virus are transiently infected but blocked the infection to a limited mucosal area by mucosal IgA or local tissue-resident T-cells, essentially conferring the protection as suggested by Wang et al. [44]. Using bronchoalveolar lavage samples, Mitsui et al. demonstrated that donors with a history of both infection and vaccination have more airway mucosal SARS-CoV-2 antibodies and memory B-cells than those only vaccinated, suggesting that peripheral vaccination alone fails to induce durable lung mucosal immunity against SARS-CoV-2 [51]. Thus, as in the case of influenza vaccine [52], mucosal priming must be considered.

During the SARS-CoV-2 pandemic, we have gained a lot of knowledge about how the human body develops immunity against acute virus infections. This information is crucial for us to prepare for any future emergence of unknown viruses. Nowadays, vaccines can be developed quickly once the genetic structure of a virus is determined. However, current vaccines are not efficient in blocking upper respiratory mucosal infections and they do not provide long-lasting protection. Therefore, we need to put in more effort to develop highly effective and long-lasting vaccines. We may also need better ways to predict the emergence of new pathogens.

## Figures and Tables

**Figure 1 viruses-16-00446-f001:**
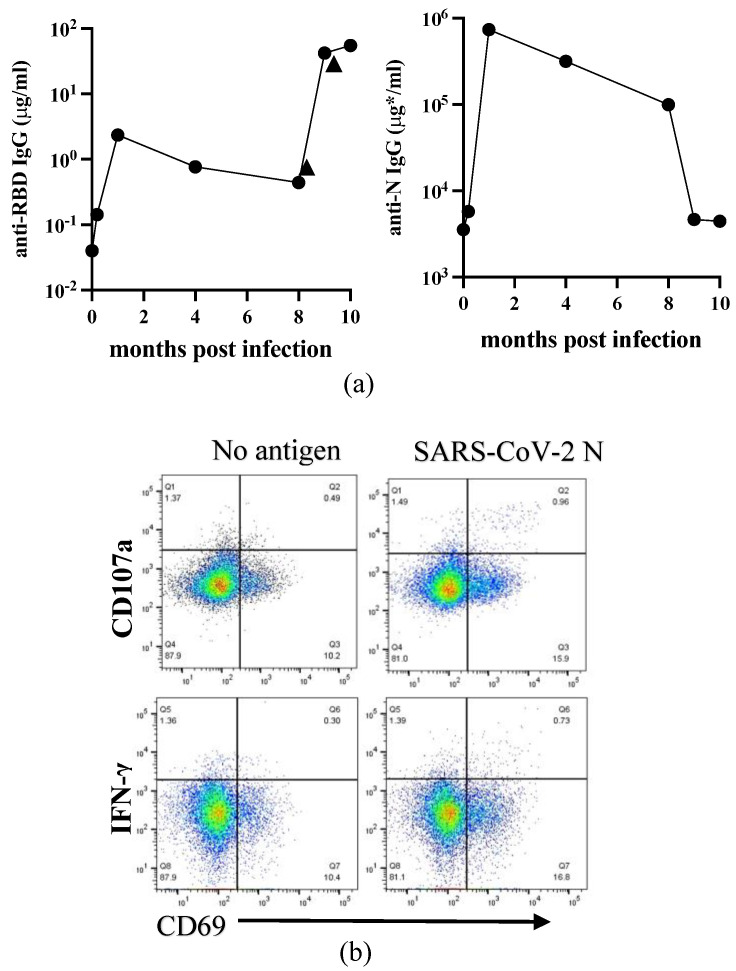
Time course of immune responses to SARS-CoV-2 in one infected individual during the early phase of the pandemic. (**a**) The levels of serum anti-RBD and anti-N IgG before and after SARS-CoV-2 infection are depicted in the left and right panels, respectively. The vertical and horizontal axes represent the amount of IgG and the time since infection. In the anti-N IgG panel, the amount of IgG was determined based on the amount of IgG purified from this individual’s serum at peak level (μg *). Arrowheads indicate two COVID-19 mRNA vaccinations. (**b**) PBMCs were either unstimulated (none, open circles) or stimulated with N protein (N, closed circles) overnight and analyzed with flow cytometry. Live CD3^+^CD8^+^ T-cell population was gated, and the frequencies of CD69^+^IFN-γ^+^ (left) or CD69^+^CD107a^+^ cells were depicted. The representative flow cytometer results one month after infection were shown. (**c**) Frequencies of CD69^+^IFN-γ^+^ (left) or CD69^+^CD107a^+^ cells at all time points were plotted.

**Figure 2 viruses-16-00446-f002:**
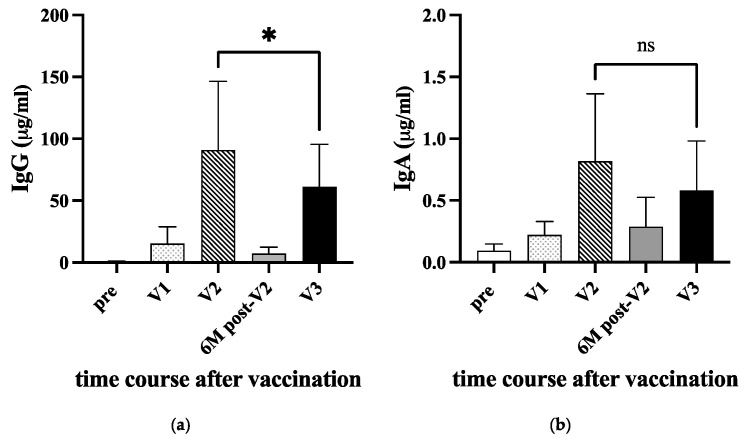
Anti-RBD IgG and IgA antibodies in sera after vaccinations. Twenty-two donors were immunized either with BNT162b2 or mRNA-1273 COVID-19 vaccine in 2021 (see Table 1). The sera were collected before immunization (pre), 10 days after priming (V1), 10–14 days after the second booster (V2), 6 months after the second (6M post-V2), and 10–14 days after 3rd immunization (V3). The amount of anti-RBD IgG (**a**) and IgA (**b**) were measured with ELISA. A *p*-value (* <0.05) and standard deviation (bars) are shown. ns: not significant.

**Figure 3 viruses-16-00446-f003:**
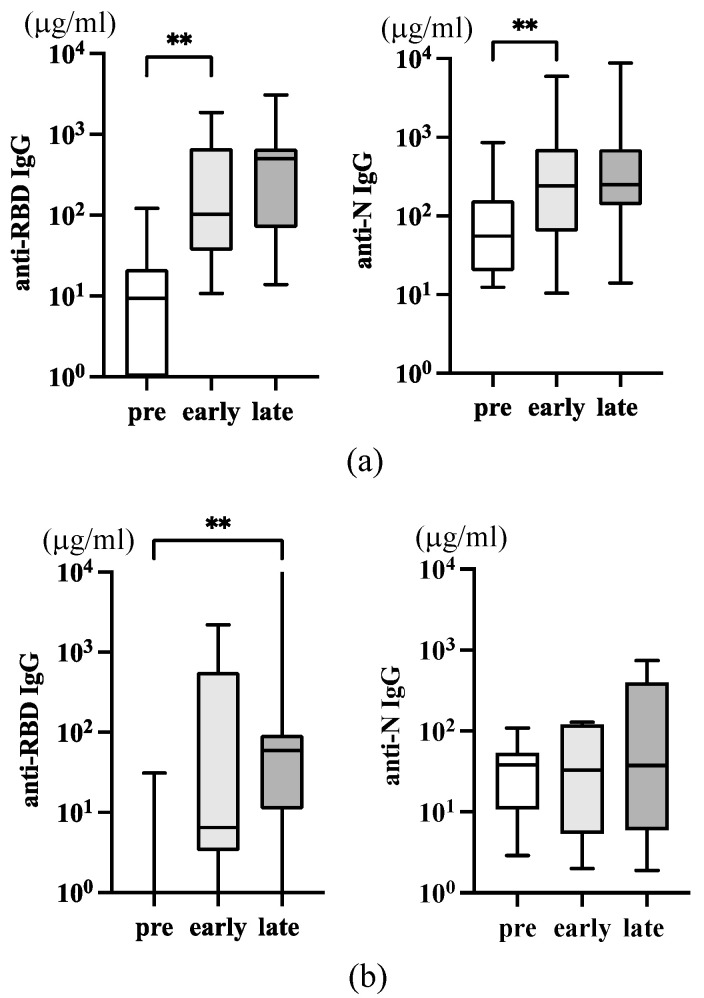
Serum anti-SARS-CoV-2 IgG antibodies in infected and close contacts. (**a**) The level of serum IgG in infected individuals (n = 17, Table 2) is depicted. The left panel is anti-RBD IgG, and the right panel is anti-N IgG. (**b**) The level of serum IgG in asymptomatic close contacts with infected individuals (n = 10, Table 3) was depicted. The *p*-values (** <0.01), mean (horizontal), and standard deviation (vertical) bars are depicted.

**Figure 4 viruses-16-00446-f004:**
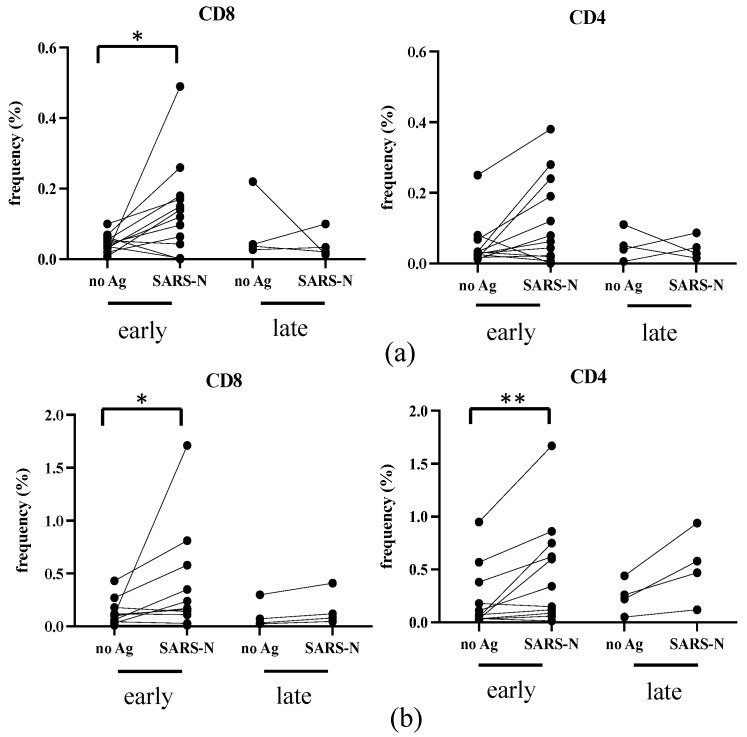
N-specific T-cell responses in infection. The PBMCs were either unstimulated or stimulated with N protein overnight and analyzed with flow cytometry. The frequencies of IFN-γ^+^ (**a**) and CD107a^+^ (**b**) in CD3^+^CD8^+^ (left panel) and CD3^+^ non-CD8^+^ T-cells (right panel) with activated phenotype (CD69^+^) are shown. early: 10–14 days after infection, late: more than 3 months after infection. The donors are indicated in Table 2.

**Figure 5 viruses-16-00446-f005:**
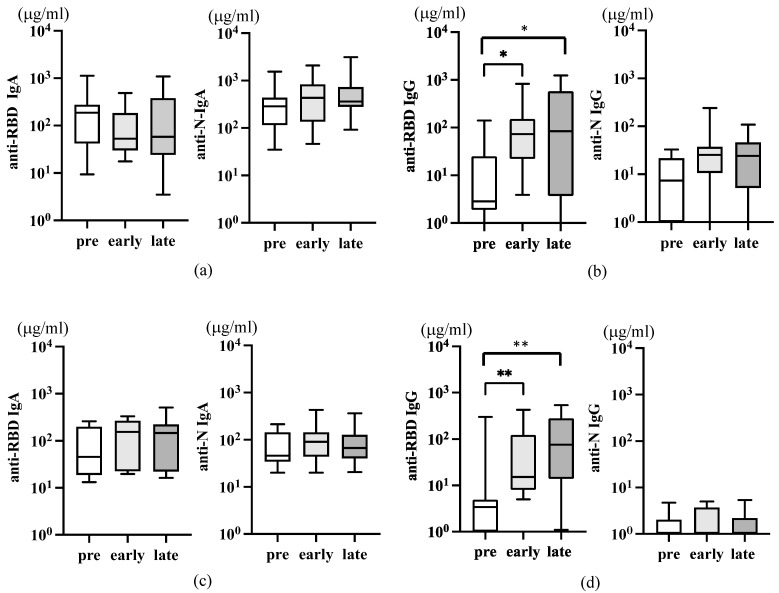
Saliva anti-SARS-CoV-2 IgA antibodies after infection and close contact individuals. The amounts of anti-RBD ((**a**–**d**), left panels) and anti-N ((**a**–**d**), right panels), IgA (**a**,**c**), and IgG (**b**,**d**) in the saliva of the same individuals described in Figure 3 were analyzed. The *p*-values (* <0.05 and ** <0.01), mean (horizontal), and standard deviation (vertical) bars are depicted.

**Figure 6 viruses-16-00446-f006:**
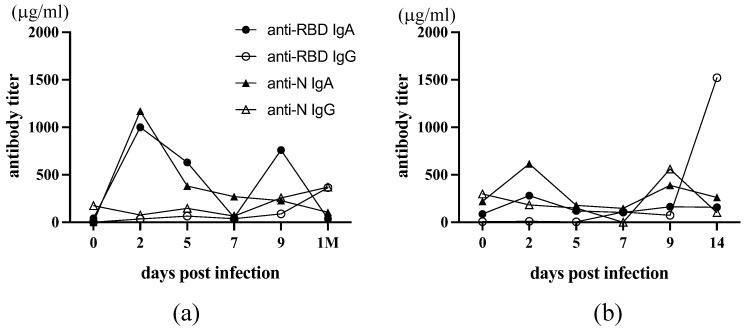
Early kinetics of saliva antibody responses after infection. The saliva samples from two infected family members were collected consecutively starting on day 0 of the SARS-COV-2 infection. (**a**) The first infected individual, (**b**) a sister infected from (**a**). 1 M: 1 month post infection. The levels of antibodies are depicted. The closed circle is anti-RBD IgA, the open circle is anti-RBD IgG, the closed triangles are anti-N IgA, and the open triangles are anti-N IgG.

**Table 1 viruses-16-00446-t001:** Information of donors who received COVID-19 vaccines in 2021.

Total number	22	
age	20–68	
sex	male	11
	female	11
vaccination 2 doses	BNT162b2	7
	mRNA-1273	15
3 doses		15

**Table 2 viruses-16-00446-t002:** Characteristics of Infected individuals.

ID	Infection in 2022	Vaccination (Times)	Month after Last Vaccination	T-Cell Analysis (PBMCs) ^#^
CV26	Feb	2	5	late
CV27	Mar	2	6	late
CV28	Mar	2	5	late
C22-1	Apr	2	6	early
C22-2	Apr	2	6.5	NA
C22-3	Apr (2nd)	2	4	late
C22-8	Jul	2	6	NA
C22-9	Jul	3	1	early
C22-10	Jul	3	2	early
C22-11	Jul	3	2.5	early
C22-13	Jul	3	3.5	NA
C22-14	Aug	3	1.5	early
C22-15	Aug	3	5	NA
C22-16	Aug	3	3	NA
C22-21	Sep	3	4.5	early
C22-22	Aug	3	4	early
C22-23	Aug	3	3	NA

Reinfection:1st infection occurred in July 2021. ^#^ NA denotes data not available, early refers to approximately 14 days after infection, and late refers to approximately 3 months after infection.

**Table 3 viruses-16-00446-t003:** Characteristics of asymptomatic close contacts.

ID	PCR Test	Contacts in 2022	Vaccination (Times)	Month after Last Vaccination
CV29	negative	Apr	2	6.5
C22-5	negative	May	2	7
C22-6	negative	May	2	7
C22-7	negative	May	2	9
C22-12	negative	Jul	3	3.5
C22-17	ND: Ag test	Aug	3	2.5
C22-18	ND: Ag test	Aug	3	3
C22-19	ND: Ag test	Aug	3	3
C22-20	ND: Ag test	Aug	2	10
C22-24	ND: Ag test	Aug	3	5

ND denotes no PCR test carried out. Instead, the Ag test was negative.

## Data Availability

The raw data (excel sheets) of this article used for figures will be made available by the authors on request to the corresponding author.

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
