# Peer review of "SARS-CoV-2-Specific Immune Responses in Vaccination and Infection during the Pandemic in 2020–2022"

_viruses, 2024, doi:10.3390/v16030446_

Round 1

Reviewer 1 Report

Comments and Suggestions for Authors

Reviewer’s Comments - Manuscript Viruses 2878000 – SarsCov2 specific immune responses I vaccination and infection during the pandemic in 2020-2022by W. Inoue et al.

In the manuscript the Authors evaluated humoral (RBD and N-specific IgG and IgA in blood and saliva) and cellular (in blood N specific CD69-INFgamma and CD69-CD107) responses during 2020-2022 in vaccinated and infected subjects and in some of their close family members contacts who remained negative (PCR and Ag test).

As a major comment, I think the organization of the manuscript needs to be revised to better focus the aim/message and rationale of the work, including the study design (i.e with a scheme of the study design and a timeline schedule) and the results. In my opinion, a main limitation seems to be the small number of subjects (i.e. one subject analysed in Figure 1; only two subjects in Figure 6) and the lack of a control group of uninfected vaccinated subject. I also believe that the manuscript should be reviewed by a native English speaker to make the reading easier.

-Line 163: I believe that “were” should be deleted: i.e ..:IgA antibodies served…. (not were served)

-Please check all over the manuscript that symbols (indicating micron, microliters

or gamma) are correct (as en example please see line 187, 189, 192 etc).

-Line 223: delete fever (repeated twice)

-Line 235: delete were stimulated since in line 234 it is written We stimulated …

-Line 247: leged to figure 1 please correct in one infected individual (instead of infected individuals). It is misleading

-Figure 1: panel b: I guess that in x-axis it should be CD69 (not CD6)

Table 2:

-Figure 2: it is not clear who has received BNT162b” and mRNA.1273 (lines 269-272 it is stated that there is no difference).

-for analysis of cellular responses several peptides of Sars-Cov2 Spike and N proteins are available that could be probably used to better dissect the cellular responses and measure not only responses against the N protein

-section 3.5 : line 372: please check figure number (i.e fig 5b should be 5c?). The same at lines 374 and 375 (i.e fig 5b should be 5c?)

-lines 365-367: it is not clear what are “ordinary individuals” and why it was impossible for them to collect saliva samples ….

-figure 6: delete day for each day number since the legend already indicate that in the x axis the numbers refer to days post infection

- line 430: perhaps in this discussion it should be clarified what “recent technology” are

-Lines 433-435: to me seems a repletion of the previous concept

Author Response

We appreciate very much for your helpful comments and suggestions. We revised our manuscript. Please find the corrections and our response to each comment.

In order to address this, I have prepared a supplementary figure as your suggestion to make this study more comprehensive. Please find the supplementary figure in the Supplementary material.

Reviewer 2 Report

Comments and Suggestions for Authors

Dear authors,

Thank you for this paper, which is very interesting despite the low number included. I have only 2 minor comments:

- page 5, line 235, there is certainly a mistake in the sentence (We stimulated PMBC....)

- Some units have problems ("@" instead of special characters)

Author Response

Thank you for your encouraging comments. We have made the necessary corrections to the grammar and symbols you pointed out.

Reviewer 3 Report

Comments and Suggestions for Authors

General Comments:

This study involved the analysis of the IgG and IgA antibodies against the Spike receptor-binding domain (RBD) and the Nucleocapsid protein in COVID-19 recovered and also in COVID-19 vaccinated individuals using serum and saliva samples. Unfortunately, there was no data provided on antibody responses to the full Spike protein and other SARS-CoV-2 proteins. Nucleocapsid antibody response is not a reliable marker of previous infection with SARS-CoV-2 based on my own experience and data in the scientific literature, although it is commonly used for this purpose.

A relatively small number of students were examined in this study: One originally identified in 2020 with mild COVID-19, 22 students tested before and following vaccination in June-August of 2021, and in 2022, 17 infected students with Omicron variants and 10 PCR-negative students that did not have COVID-19 symptoms, but were vaccinated. In some instances, such as time courses of antibody production in saliva based on only two people, the data is very limited in the number of subjects. Large differences in antibody responses are typical from one person to another.

The conclusions are generally supported by the data, although presentation of the data can be improved. It is difficult to compare the results with unvaccinated students, so specific comments in this respect need to be carefully made. I suspect that almost all of the students had prior exposures to SARS-CoV-2 before they were vaccinated.

Specific Comments:

1.      Abstract – “One patient who had been infected in 2020 developed serum RBD and N-specific IgG antibodies, but declined 8 months later, then mRNA vaccination in 2021 produced a higher level of anti-RBD IgG than natural infection.” This statement is hardly surprising as antibodies levels will normally subside after the body is not challenged with a pathogen.

2.      Abstract – “In the vaccination of naïve individuals, vaccines induced anti-RBD IgG, but it declined after 6 months. A third vaccination boosted the IgG level again, albeit to a lower level than after the second.”  Again, this is not a surprising result, with more than two COVID-19 inoculations, there is a shift to the production from IgG1 and IgG3 to IgG2 and IgG4 class antibodies, which confer immune tolerance. [Uversky, V.N., Redwan, E.M., Makis, W., Rubio-Casillas, A. (2023) IgG4 antibodies induced by repeated vaccination may generate immune tolerance to the SARS-CoV-2 spike protein. Vaccines (Basel). 11(5):991. doi:10.3390/vaccines11050991 ; Irrgang, P., Gerling, J., Kocher, K., Lapuente, D., Steininger, P., et al. (2023) Class switch toward noninflammatory, spike-specific IgG4 antibodies after repeated SARS-CoV-2 mRNA vaccination. Sci Immunol. 8(79):eade2798. doi:10.1126/sciimmunol.ade2798  ; Kiszel, P., S.k, P., Mikl.s, J., Kajd.csi, E., Sinkovits, G., et al. (2023) Class switch towards spike protein-specific IgG4 antibodies after SARS-CoV-2 mRNA vaccination depends on prior infection history. Sci Rep. 13(1):13166. doi:10.1038/s41598-023]

3.      Abstract – “In saliva, SARS-CoV-2-specific IgG but not IgA was detected.” It is ambiguous whether this statement refers to a SARS-CoV-2 virul infection or after COVID-19 vaccination. With COVID-19 vaccination into the arm, an IgG response would be expected. IgA and IgM  levels in response to infections in the nasopharyngeal cavity is usually higher than IgG. Thomas et al. (2023) have noted that salivary Spike IgA was poorly correlated with serum, indicating an oral mucosal response whereas salivary Spike IgG responses were predictive of those in serum. [Thomas, A.C., Oliver, E., Baum, H.E. et al. (2023) Evaluation and deployment of isotype-specific salivary antibody assays for detecting previous SARS-CoV-2 infection in children and adults. Commun Med3:37. https://doi.org/10.1038/s43856-023-00264-2] The ramification of this is that the protective response to SARS-CoV-2 infection in the upper airway is likely to be poorer from the COVID-19 vaccines than from a natural infection, where muscosal IgM and IgA antibodies are expected to predominate.

4.      Line 50 – change “thus rapidly developing severe pneumonia.” to “with the potential for development of severe pneumonia in especially the elderly and those with comorbidities.” The vast majority of people that have been infected and even those that get COVID-19 do not develop pneumonia.

5.      Line 56 – I believe that “P.1” is actually a variant too. It is better to refer to these viruses arising from the Wuhan SARS-CoV-2 viruses. Even the Wuhan versions may not be original SARS-CoV-2 virus as there are three variants of this too.

6.      Line 58 and 59 - At the end of 2021, the Omicron (B.1.1.529 line-58 age) variant emerged in South Africa and became a significant variant worldwide.” The first Omicron variants did not originate in South Africa but was first reported there. It is believed to have arisen in Europe or possibly Puerto Rico. [Tanaka, A., Miyazawa, T. (2023) Unnaturalness in the evolution process of the SARS-CoV-2 variants and the possibility of deliberate natural selection. Zendo. doi:10.5281/zendo.8216373]

7.      Lines 62-64 – The “https://www.who.int/news/item/16-03-2023-statement-on-the-up-date-of-who-s-working-definitions-and-tracking-system-for-sars-cov-2-variants-of-concern-and-variants-of-interest” url does not seem to work.

8.      Line 67 – “The rapid development of novel vaccine technology has helped us achieve herd immunity against SARS-CoV-2 infection in the general population.”  I do not think this statement is correct. Most cases of COVID-19 in the last two years have been in people that are double or triple vaccinated against SARS-CoV-2. The reduction in COVID-19 cases is more likely due to reduced virulence of the omicron variants and the acquisition of natural immunity from actual infection with the SARS-CoV-2 virus.

9.      Line 69 – Contrary to this statement, it is feasible that the COVID-19 genetic vaccines have actually prolonged the COVID-19 pandemic. The repeated booster injections appear to increase the chances of getting COVID-19. The authors may wish to examine the recent study from the Cleveland Clinic that showed that more vaccination progressively increased the number of healthcare works were got COVID-19 over a 6 month period starting in the summer of 2022. [Shrestha, N.K., Burke, P.C., Nowacki, A.S., Simon J.F., Hagen, A., Gordon, S.M. (2023) Effectiveness of the coronavirus disease 2019 bivalent vaccine. Open Forum Infect Dis. 10(6):ofad209. doi:10.1093/ofid/ofad209

10.   Line 85 – “Thus, these findings indicate that re-infection frequently occurs after vaccinations and even after natural infections, probably associated with continuous virus mutations.”  This seems highly unlikely, since even the Omicron variants have >96% amino acid identity with original Wuhan Spike protein and antibodies are generated against many different regions in Spike protein. The idea of loss of immunity is based on the observation that mutations within or near the receptor binding domain reduce the effectiveness of “neutralizing antibodies.” However, the vast majority of antibodies generated against the Spike protein in SARS-CoV-2 infected individuals and vaccinated individuals do not target the RDB, which is actually poorly immunogenic. As the authors should know, the binding of antibodies to any external protein on the surface of a virus enable recognition by innate immune cells to attack that virus. Likewise, most T-cell responses are unlikely to depend on recognition of the RBD region of the Spike protein.

11.   Lines 94 and 95 – “The students in our university were relatively free from COVID-19 during several waves of epidemics in 2020 and 2021, probably by avoiding close contact among students and staff.” COVID-19 is usually asymptomatic in children, and probably in most university students as well. Without proper serological testing with sensitive assays, this would be hard to know and is speculation. The high rates of COVID-19 cases in university students probably reflected waning immunity from earlier infections with SARS-CoV-2 after 1 to 2 years, and the increased infectiousness of the Omicron variants. However, most university students were mildly sick, likely a combination of previous natural immunity and COVID-19 vaccination. The high rate of COVID-19 vaccination in Japan could have actually increased susceptibility to SARS-CoV-2 infection, especially when these inoculations were performed during a peak of infections when the level of the virus was high in the environment. This can be explained by the hyper-production of the Spike protein in the cells of the bodies of vaccine recipients soon after vaccination. This diverts the antibodies and T-cells specific for the Spike protein to mediate inflammatory attacks on the Spike-producing cells, including the recruitment of innate immune cells. As a consequence, the mobile innate and adaptive immunity that might have been available to take on a SARS-CoV-2 infection in the nasopharyngeal space is diverted further into the body and permits less-restricted establishment of immune protection against the virus. Immediately following COVID-19 vaccination, there is actually an increase in COVID-19 cases, which is why most health authorities consider people unvaccinated until two to three weeks after their first vaccination, and partially vaccinated until two weeks after their third vaccination.

12.   Line 113 – It is unclear if whole blood samples were frozen, or if the serum was first isolated and this was frozen. If whole blood cells were frozen, then erythrocyte proteins may have been released during the freeze-thaw process. Some erythrocyte proteins can interfere with the binding of antibodies and compete with target antigens. This can lower the sensitivity of detection of antibodies. Was this controlled for in this study by comparison with serum only samples?

13.   Line 128 – Saliva samples are tricky to deal with, because the amount of water in the saliva specimen can vary markedly from person to person. How was the concentration of saliva proteins controlled for? Was a reference protein used to standardize the concentrations for comparison, since the total immunoglobulin levels would be expected to change with infection or vaccination?

14.   Line 137 – It appears that the SARS-CoV-1 Nucleocapsid protein rather than SARS-CoV-2 Nucleocapsid protein was used as the antigen in the serological tests performed in this study. The justification given is that it was 90% identical in amino acid sequence. Earlier, the authors pointed out in the Introduction that antibody recognition of the Spike protein from the Wuhan strain was markedly diminished towards the Spike proteins of Omicron variants, but these are greater than 96% in amino acid identity. Thus, it may not be surprising that natural immunity from a previous SARS-CoV-2 infection might not be picked up with this particular antigen that was used in the assay. Since the authors did generate a recombinant SARS-CoV-2 Nucleocapsid protein in their lab for the T-cell studies, was this used to validate the usage of the SARS-CoV-1 Nucleocapsid in the serological antibody testing work?

15.   Line 143 – Only a small portion of the Spike protein (from the Wuhan strain) was used an antigen to test the COVID-19-vaccine response. This would be subject to the same caveats as “neutralization assays” mentioned above. It would have been better to use whole Spike protein as an antigen to get a fuller sense of the anti-Spike antibody levels in the study participants. With changes in the RDB sequences from mutation in SARS-CoV-2 Spike proteins in the variants that predominated, which largely accounted for increased binding to ACE2 receptor and higher infectivity, this would introduce more issues with the sensitivity of the assay for anti-Spike antibody detection.

16.   Line 153 – The authors appeared to produce their own IgG- and IgA-specific Spike RDB antibodies. What were the class isotypes of these immunoglobulins?

17.   Lines 171 and 194 as well as elsewhere – change “hrs” to “hr”

18.   Lines 187, 189 and 192 – correct “@/” to what it should be. This is a problem that reoccurs later in the pdf copy of this manuscript.

19.   Lines 191 and 192, change “5 μg/mL and two μM, respectively.” to “5 μg/mL and 2 μM, respectively.” 

20.   Lines 220 to 233 - The very rapid drop in anti-Nucleocapsid antibody levels in one month during the COVID-19 vaccination period is surprising. Is the natural immunity against this SARS-CoV-2 protein being suppressed? While a reduction of anti-Nucleocapsid antibodies is not particularly a concern (since the protein is shielded inside the virus particle from the immune system), it would be interesting to know what happened to the antibody levels against other SARS-CoV-2 proteins such as the Membrane protein. One concern about this experiment is that IgG levels were analyzed. Since the patient had a minor response to infection that was a slight fever, it would seem likely that the antibody response should be less robust. The response would probably be much higher in someone who was really sick.

21.   Figure 1 - In the y-xis labels used in the top two panels, why is a small “m” used? Is this supposed to be “µ”? In the lower panels, a small “g” appears after “IFN”. I presume this is gamma. Since the panel above has “IFN-g”, the lower left panel should be changed to be consistent. I am a little confused by how the concentration values for the anti-Nucleocapsid antibodies was determined.

22.   Lines 236, 240 and 241 show nonstandard symbols on the pdf that are not recognizable.

23.   Line 247 – “Immune” should not be capitalized. The title infers that testing was done in individuals, where it was done for only one person in this figure. It should be mentioned in the two upper panels that serum samples were analyzed.

24.   Line 261 – What was the difference in time between the first inoculation (V1) and the second inoculation (10 days before V2)? Likewise, what was the difference in time between the second inoculation and the third inoculation (10-14 days before V3)?

25.   Line 265 – Change to “Fig.2a” to “Fig. 2a

26.   Table 1 – The last column should have the numbers aligned.

27.   Line 259 to 271 – In the 22 volunteers, it is unclear if they already had COVID-19 previously. The detectable IgA antibody level in the “pre” average value indicates that some of these participants may have already been infected with the virus, and the antibody levels had waned since infection, which could have been a year or more earlier. If any of these individuals had COVID-19 symptoms or had tested positive by PCR for SARS-CoV-2, their data should be separated out and presented as a second group. This appears to be true for at least student Case 22-3. In Figure 2, it is apparent that there were huge differences in the antibody responses amongst the students based on the error bars. While not significant in the IgA case, but was in the IgG case, it appears that the third vaccine shot did not generate as strong an antibody response. This is consistent with a degree of tolerization induced with the third shot as compared to the second shot. Why was measurement of their serum IgA and serum IgM antibodies for the Nucleocapsid protein not also provided during this period and shown?

28.   Table 2. The average number of months for a breakthrough COVID-19 infection amongst the double vaccinated can be calculated to be 5.5 months. In those that were triple vaccinated the average was 3 months. It seems that the three doses of vaccines were poorly protective during the Omicron period. Case C22-3 would effectively be like a person that had three vaccine shots, since they had previously been infected with SARS-CoV-2 before their two vaccinations. It is noteworthy that they had only 4 months of protection since their second vaccination before onset of COVID-19, which was the shortest period for a double vaccinated person before breakthrough COVID-19 in this group. All of these results point towards reduced protection against COVID-19 with increasing vaccination. This is consistent with the findings from the Cleveland Clinic with repeated booster injections of COVID-19 vaccines in healthcare workers. [Shrestha, N.K., Burke, P.C., Nowacki, A.S., Simon J.F., Hagen, A., Gordon, S.M. (2023) Effectiveness of the coronavirus disease 2019 bivalent vaccine. Open Forum Infect Dis. 10(6):ofad209. doi:10.1093/ofid/ofad209]

29.   Lines 302-307 – In Figure 3, the infection with Omicron clearly increased the detection of anti-Spike RBD domain antibodies. However, the assays used employed Wuhan Spike RBD as the antigen. These data clearly indicate that there must be sufficient overlap of the Wuhan and Omicron RBD sequences to induce production of more antibodies against this region. And yet, these vaccinated individuals still got sick with COVID-19. To understand this, it would be nice to see if there was a shift in the IgG antibodies to the IgG4 class.

30.   Figure 3a and 3b – It is evident that all of the students that got COVID-19 after their vaccination had previously been infected with SARS-CoV-2 based on appreciable anti-Nucleocapsid antibody levels prior to their subsequent breakthrough COVID-19. Likewise, the close contacts were probably all infected previously. The PCR test only demonstrates that the close contacts were not infected to a measureable degree at the time that the students who were sick with COVID-19. As the authors have correctly interpreted these finding, the close contacts had sufficient immunity to protect them from sickness and were probably naturally boosted by exposure to the SARS-CoV-2 virus. Since the anti-Nucleocapsid and anti-RBD antibody levels were similar in the two groups, other antibodies or other factors seem to be more important than vaccination status. This may include the time from when they were originally infected with virus and well as differences in their innate and T-cell immunity.

31.   Figures 2, 3 and 5 could be improved by showing the individuals data points in the bar graphs as this would give a better sense of the distribution in results for the individuals within each group.

32.   Lines 324 and 326 – Change “was depicted.” to “is depicted.”

33.   Lines 329 and 330 – It is hard to see how T-cell responses against Nucleocapsid protein would be protective, although they would be generated following an infection along with anti-Nucleocapsid antibodies. T-cells could recognize virus infected cells as the virus particles emerge from cells. However, the Nucleocapsid protein would not be visible from outside of the cell, even during the blebbing of the virus.

34.   Lines 328 to 352 – The results for measurement of T cell activation following SARS-CoV-2 infection are not surprising, although the sample size is small and only a couple of time points are presented. It would have been much more interesting to see the T-cell responses to COVID-19 vaccination.

35.   Lines 358-371 – The delay in detection of IgA antibodies after infection is problematic since the half-life of these type of antibodies is only about 5-6 days, whereas IgG has a half-life of over 3 weeks. Therefore, it will be easier to see these antibodies following recovery from infection at a later time than IgA antibodies. The university regulations that the authors faced was problematic, and the results are not surprising if their assay was not sensitive-enough (and against SARS-CoV-1 Nucleocapsid protein rather than SARS-CoV-2 Nucleocapsid protein). I have worked with saliva samples myself even months after SARS-CoV-2 infections, and I have been able to detect anti-Nucleocapsid antibodies in many of my test subjects. However, I have also learned that about half of the COVID-19 case individuals that I performed serum antibody measurements a few months after their infection in the pre-Omicron phase fail to produce detectable anti-Nucleocapsid antibodies, but clearly had antibodies against the Spike, Membrane and several of the nonstructural proteins of the SARS-CoV-2 virus even after two years in individuals that have not been vaccinated. With the more infectious Omicron variants, it appears to be easier to detect anti-Nucleocapsid antibodies, even with less sensitive serological tests.

36.   Lines 370 to 371 – Inspection of Figure 5 does seem to show appreciable levels of IgA in the serum samples, but these are not increased with a subsequent infection.

37.   Lines 372 to 378 – The saliva data for anti-RBD and anti-Nucleocapsid IgA levels prior to the latest infection indicates that these individuals had likely been previously infected with SARS-CoV-2.

38.   Line 403 – In Figure 6, the difference in the right and left panels should be explained. I surmise that each panel corresponds to a different person. “1M” in the figure should be defined. Is this 1 month?

39.   Line 419-421 – In this study, unvaccinated people that clearly could not have been infected with SARS-CoV-2 before were not analyzed, except for a doctor that had a mild COVID-19 in 2020. Therefore, the level of antibody response to a SARS-CoV-2 infection that leads to more pronounced COVID-19 without vaccination is largely unexplored in this study. At 8 months after infection, the antibody levels against the Spike RBD region and Nucleocapsid protein would have substantially declined, so it is not surprising that vaccination with tens of trillions of lipid nanoparticles that contain 5 to 10 copies of a genetic engineering mRNA with N1-pseudouridine for increased stability and greater production of the Spike protein in body cells would elicit a strong immune response. I doubt that such levels of viral antigen could be achieved naturally with a virus without producing severe illness.

40.   Lines 431 to 435 – It seems to me that the authors data reveal issues with booster vaccination with COVID-19 vaccines, which if effective provides only a temporary protection, which may have negative efficacy subsequently. The statement concludes with “especially for individuals who have previously been infected with the virus.”  However, if person has had COVID-19 and full recovered due to natural immunity against the whole virus that involves a mucosal antibody response, and the SARS-CoV-2 had mutated to less virulent form, I cannot understand why that person would be disadvantaged relative to a person that has only been vaccinated against one of the virus’s proteins, with primarily an IgG response that is less lasting.

41.   Lines 437-439 – There is only data provided from one person where it is unlikely that the person had not previously been infected by SARS-CoV-2. Since everyone else was tested in the summer of 2021 or later, and they could still have been infected with SARS-CoV-2, but were asymptomatic at the time.

42.   Lines 447 and 448 – The apparent increase in vaccine effectiveness in preventing COVID-19 after a third dose could simply be because more benign SARS-CoV-2 variants predominated and there was already mounting natural immunity in the population.

43.   Line 456 – The cited study (ref. 34) from the U.K. involves self-reporting in an unblinded study by the participants and the researchers, and corresponds to pre-Omicron SARS-CoV-2 infections. Out of more than 1.5 million initial participants, the final number was filtered into 6 groups with a total of around 33,000 participants, of whom typically 55% or more were vaccinated with the AstraZeneca adenovirus vaccine. COVID-19 outcomes were poorer in the vaccinated group after their first dose than in the unvaccinated group if the participant was obese or elderly, which correspond to higher risk groups for COVID-19. Sneezing was also higher in the vaccinated group following initial vaccination.

44.   References – There is inconsistency in the use of capital letters in the words used for titles of articles.

Comments on the Quality of English Language

There is some editing that I have listed above. Overall, the English is good and relatively easy to follow.

Author Response

We sincerely appreciate your valuable and meticulous feedback on our work. We have carefully reviewed your suggestions and comments and have made necessary changes based on them.

Round 2

Reviewer 1 Report

Comments and Suggestions for Authors

Eventhough my previous comment regarding the low number of subjects enrolled is still valid, I appreciate that the autors have replied to my comments and revised the paper accordingly as much as possible. 

Author Response

Thank you for your comment again. We have modified the Abstract and Discussion to address the concern regarding the small sample size in this study.(blue colored)

Reviewer 3 Report

Comments and Suggestions for Authors

Inoue et al. have provided a length response to my critique of their manuscript, and have endeavoured to address many of the points that I have raised. I will deal with what I think are the outstanding issues.

General Comments

1.     The authors are convinced that anti-Spike receptor binding domain (RBD) antibody measurements are sufficient to assess COVID-19 vaccine-induced as well as natural immunity from SARS-CoV-2 infection. They cite other publications where this has been done as well. They also maintain that antibodies developed against SARS-CoV-1 Spike-RBD do not react with SARS-CoV-2 RBD. The RBD region of Wuhan SARS-CoV-2 Spike RBD spans 223 residues (aa # 319 to 541) in a protein of about 1273 amino acids (aa), which is about 17.5% of the total structure. In my own research, I have identified SARS-CoV-2 patients that clearly have antibodies that cross-react with SARS-CoV-1 RBD sequences in aa 398-412 (7/30 patients), aa 417-431 (14/30 patients), aa 426-430 (9/30 patients), aa 433-447 (13/30 patients), aa 437-452 (11/30 patients), aa 463-476 (29/30 patients), and aa 491-505 (29/30 patients). The peptide amino acid numbering corresponds the Wuhan SARS-CoV-2 Spike protein in terms of position in the protein. Although it is highly unlikely that the students tested had previously had SARS-CoV-1, there is a high probability that they have been exposed to  coronaviruses in the past that may have had the virulence of the common cold, but was considered an influenza-like illness. The fact that the COVID-19 patient set of 30 participants had such cross-reactivity with a coronaviruses sequences from over 20 year ago in my studies indicates that any tests with the RBD region of SARS-CoV-2 cannot be taken as specific.

2.     The authors indicated that it was not convenient for them to use the whole SARS-CoV-2 protein as the immunogen. They indicated that it would have been more laborious, but from a labour standpoint, I would not expect much difference in the work needed to make the recombinant full length protein. The wild-type recombinant SARS-CoV-2 Spike protein is commercially available from many sources. Having the full Spike protein would have improved the sensitivity of the serological test to avoid false-negatives. Nonetheless, the specificity for distinguishing from other coronaviruses would still be an issue as with using just the RBD.

3.      As I mentioned in my previous review, the vast majority of anti-Spike antibodies target sequences outside of the RBD, and the RBD is actually poorly immunogenic.

4.     The authors used SARS-CoV-1 Nucleocapsid antibody as an antigen for the serological test. This was based on high conservation in amino acid sequence between the Nucleocapsid proteins in SARS-CoV-1 and SARS-CoV-2, which share an identity of 89.3% across 420 amino acids. This is better in amino acid identity when compared to the nucleocapsid identity between SARS-CoV-2 and Bat CoV/279/2005 [UniProt Q0Q468], which is 78.2%, but there are very large tracks of complete amino acid identity between these viral proteins, which would strongly support potential cross-reactivities. Therefore, when using the SARS-CoV-1 nucleocapsid as an antigen for SARS-CoV-2 antibody detection, it is feasible that it could be able to pick up other beta-coronaviruses. In the publication by Leung et al. (2004; J. Infect. Dis. 190(2):379-386) referred to by the authors, in addition to IgG responses, they also noted IgM responses were detectable in the SARS-CoV-1 patients, whereas IgA was not specifically tested. It was also noted in this study that 21% of the tested SARS-CoV-1 patients failed to produce anti-Nucleocapsid antibodies and 60% did not have detectable Spike antibodies. This supports my earlier assertion in my previous review that Nucleocapsid antibodies are often not detectable in SARS-CoV-2 patients and should not be used alone as proof that a person has not previously been infected by SARS-CoV-2.

5.     The authors state “However, unlike large-scale studies, our study allows us to evaluate how the SARS-CoV-2 pandemic affects an individual's human immunity in response to a novel virus, and we can follow the outcome over the study period in our small community.” I fail to see how this can be true. Surely with more people tested, the greater the opportunity to identify differences in antibody responses.

6.     The authors have argued that the students in their study did not have previous infections with SARS-CoV-2 before mid-2021, when they were vaccinated. Assuming that strict measures taken by the university to restrict the spread of COVID-19 is not reassuring to discount previous infection. For example, the use of N95 masks have not been satisfactorily demonstrated to reduce infection and transmission of SARS-CoV-2. SARS-CoV-2 was already in Wuhan in November and likely October of 2019. That the virus did not widely spread into Japan almost 2 years later is highly unlikely. The more likely scenario is that the students already had sufficient innate and previous adaptive immunity from cross-reactive to coronaviruses and possibly SARS-CoV-2 itself to have antibody levels that had waned to below detectable levels by the time of the study commencement.  As described in the study by Majdoubi et al. (2021; JCI Insight. 6(8):e146316), 90% of 276 healthy adults test for Spike and Nucleocapsid antibodies in the late Spring of 2020 already had detectable antibodies. The negative results with the Japanese students is more likely due to the poor sensitivity of the serological tests that were used and the poor rate of anti-Nucleocapsid protein production with asymptomatic infections.

Specific Comments (These are numbered based on their original numbering in the authors’ response)

2. The authors state, “We expected that the third vaccination would further increase the antibodies, but it seems unlikely. We have looked into the regulation of the amount of specific IgG in our body after repeated vaccination by the same antigen.” I would expect a temporary increase in the general titre of the IgG antibodies, particularly the IgG4 class linked to tolerance. I don’t know why the authors think otherwise, and it is unclear what the authors discovered from their unidentified studies of IgG titre following repeated vaccinations.

3.  It is odd that the authors found a poor IgA response to SARS-CoV-2 infection. In my own lab, we do find appreciable IgA in patients that have recovered from COVID-19. This indicates that the anti-Spike RDB antibody test used by the authors is not very sensitive. It may be that in two donors the IgA response was high enough to just meet detection threshold.

7.     https://www.who.int/news/item/16-03-2023-statement-on-the-up-date-of-who-s-working-definitions-and-tracking-system-for-sars-cov-2-variants-of-concern-and-variants-of-interest as stated does not work. The problem seems to be the inclusion of a hyphen in “up-date”. https://www.who.int/news/item/16-03-2023-statement-on-the-update-of-who-s-working-definitions-and-tracking-system-for-sars-cov-2-variants-of-concern-and-variants-of-interest does work. This correction needs to be made.

9. The authors have now cited the Shrestha et al. (2023) publication. However, the main point was that the incidence of COVID-19 in this study increased with increasing number of COVID-19 vaccine inoculations. The authors focused on a different aspect of this publication, but ignored the real issue of poor vaccine efficacy with booster inoculations of COVID-19 vaccines in line 96-98.

10.       The authors hold (as mentioned in lines 112 and 113) that the RBD region of the Spike protein is the most important part of the virus, and antibodies against this region are neutralizing for viral entry. While neutralizing antibodies do contribute to the immune response, this is only a small part of the action of antibodies. Most viruses are “neutralized” by innate immune cells that are attracted to the virus by way of bound antibodies. The authors have tried to argue that mutations within the RBD in the Omicron variants have resulted in loss of “neutralization” by antibodies that target the RBD, and this results in vaccinated individuals becoming infected again and getting COVID-19. However, the COVID-19 vaccines were based on the Wuhan strain, and these were still found to be effective, for at least a few months, in reducing the occurrence of COVID-19 in triple vaccinated individuals when Omicron first dominated, about 5 months before the bivalent COVID-19 vaccines were available. Clearly, antibodies against the original Wuhan SARS-CoV-2 Spike protein provided protection against Omicron BA.1 and BA.2 variants. The Omicron BA.1 Spike only differs by 17 out of about 200 amino acid residues that constitute the RBD’s, with 91.5% amino acid identity.

11.       Children were likely to be asymptomatic to SARS-CoV-2 due to stronger innate immune responses and lower levels of ACE2, the receptor for SARS-CoV-2 in the nasopharyngeal regions. In general, I suspect that there were lower rates of COVID-19 and deaths in East Asian countries due to higher exposures to more benign beta-coronaviruses in general, which produced a higher level of herd immunity in these populations. This is evident in the 900-times higher rate of COVID-19 deaths per capita in the USA compared to China.

13. I take it that the amount of IgA for the Spike RBD and Nucleocapsid proteins was monitored relative to the total amount of IgA in the saliva. However, with an infection, there would probably be an increase in the total IgA level.

14. As pointed out in my original review, the whole Omicron BA.1 and Wuhan Spike proteins are about 97% identical in amino acid sequence. There are about 31 amino acid substitutions between these Spike proteins, which with a 1273 amino acid protein would be 97.6% identity (Park, S.B., Khan, M., Chiliveri, S.C. et al. SARS-CoV-2 omicron variants harbor spike protein mutations responsible for their attenuated fusogenic phenotype. Commun Biol 6, 556 (2023). https://doi.org/10.1038/s42003-023-04923-x)

14. There actually is in fact cross-reactivity of patients that have recovered from COVID-19 with alpha-coronaviruses such as HKU1 and OC43. For example, in our own studies, 10 of 30 tested patients had antibodies for HKU1, and 13 of 30 tested patients for OC43, in the region that corresponded to ATVLQLPQGTTLPKG in SARS-CoV-2 Nucleocapsid protein, with 15/30 patients testing positive. For another SARS-CoV-2 Nucleocapsid protein sequence, i.e., QQGQTVTKKSAAEAS, 16 out 30 patients showing immunoreactivity with this Wuhan sequence, but 15 out 30 COVID-19 patients had antibodies that reacted with the corresponding OC43 sequence. For yet another SARS-CoV-2 Nucleocapsid protein sequence, i.e., GAIKLDDKDPNFKDQ,7 out of 30 patients showed immunoreactivity with this Wuhan sequence, whereas 8/30 cross-reacted with the corresponding HKU1 sequence and 14/30 cross-reacted with the corresponding OC43 sequence. Therefore, if whole Nucleocapsid protein was used as the antigen in serological tests, around about a third of tested recovered SARS-CoV-2 patients would likely test positive if HKU1 or OC43 Nucleocapsid proteins were used in the serological tests instead. However, around half of the SARS-CoV-2-infected and recovered patients would fail to yield any detectable positive antibody signals with Wuhan Nucleocapsid protein as the antigen. The fact that a monoclonal antibody against SARS-CoV-1

failed to react with OC43 in the authors hands, does not preclude the possibility that a polyclonal antibody response against SARS-CoV-2 will not cross-react with OC43 Nucleocapsid protein as discussed above.

33. Fragments of Nucleocapsid protein would be presented with MHC antigens on the surface of antigen-presenting immune cells that have been phagocytosed from virus infected cells or digested virus particles. However, as I pointed out, these antibodies would not be particularly useful in the immune response, unless the cells to be attacked are virus-infected antigen-presenting immune cells.

New Comments

1.     Line 71 – I am not so sure that the introduction of the COVID-19 vaccines has substantially altered the course of the pandemic in a such a positive way, and it may have been quite detrimental for a significant portion of vaccine recipients.

2.     Lines 163 and 164 – The statement that “the N proteins and RBD region of Spike are major immunodominant proteins in SARS-CoV-2 infection” is simply untrue in my experience. In my lab, we have actually analyzed the immune responses to all 28 of the SARS-CoV-2 protein. The RBD region of the Spike protein is weakly immunogenic compared to the rest of the Spike protein. Nucleocapsid protein fails to yield appreciable signals in about half of those that have recovered from COVID-19. The Membrane protein is much more immunogenic in a wider range of patients. There is no reference cited to support this claim that the Spike RDB and Nucleocapsid are major immunodominant proteins.

3.     Line 289 and 290 – Unless these students were monitored by PCR, I don’t think you can assume that none of the volunteers remained unaffected during the vaccination study. It can be stated that the volunteers did not display any symptoms of COVID-19, and if infected with SARS-CoV-2 were asymptomatic.

Author Response

Thank you for your helpful and thought-provoking feedback. 

Upon reviewing the manuscript, we have made the necessary changes and highlighted them in blue for your convenience. We hope this revised version addresses your concerns. 
